# Mathematical Profile Test: A Preliminary Evaluation of an Online Assessment for Mathematics Skills of Children in Grades 1–6

**DOI:** 10.3390/bs10080126

**Published:** 2020-08-04

**Authors:** Giannis Karagiannakis, Marie-Pascale Noël

**Affiliations:** 1Department of Primary Education, University of Athens, 10680 Athens, Greece; 2Department of Psychology, Université Catholique de Louvain, UCLouvain, 1348 Louvain-la-Neuve, Belgium; marie-pascale.noel@uclouvain.be

**Keywords:** assessment tool, dyscalculia, mathematical learning difficulties, mathematical abilities

## Abstract

The domain of numerical cognition still lacks an assessment tool that is theoretically driven and that covers a wide range of key numerical processes with the aim of identifying the learning profiles of children with difficulties in mathematics (MD) or dyscalculia. This paper is the first presentation of an online collectively administered tool developed to meet these goals. The Mathematical Profile Test (MathPro Test) includes 18 subtests that assess numerical skills related to the core number domain or to the visual-spatial, memory or reasoning domains. The specific aim of this paper is to present the preliminary evaluation both of the sensitivity and the psychometric characteristics of the individual measures of the MathPro Test, which was administered to 622 primary school children (grades 1–6) in Belgium. Performance on the subtests increased across all grades and varied along the level of difficulty of the items, supporting the sensitivity of the test. The MathPro Test also showed satisfactory internal consistency and significant and stable correlation with a standardized test in mathematics across all grades. In particular, the achievement in mathematics was strongly associated with the performance on the subtests assessing the reasoning and the visuospatial domains throughout all school grades, whereas associations with the core number and memory tasks were found mainly in the younger children. MD children performed significantly lower than their peers; these differences in performance on the MathPro subtests also varied according to the school grades, informing us about the developmental changes of the weaknesses of children with MD. These results suggest that the MathPro Test is a very promising tool for conducting large scale research and for clinicians to sketch out the mathematical profile of children with MD or dyscalculia.

## 1. Introduction

About one third of primary school lessons deal with mathematics because ability in this domain is crucial for success in Western societies [1]. Yet, not everyone is able to master even the basics of mathematics. According to the PISA (Programme for International Student Assessment) survey, 19% of Belgian students are considered to be low achievers in mathematics (OECD, 2016). The average proportion of low-achieving 15-year-old students in mathematics in EU member countries has remained essentially the same in PISA 2015 (22.2%) in comparison to PISA 2012 (22.1%), which is an indication that the EU countries are not making sufficient progress towards reducing the proportion of low achievers in mathematics. Children who subsequently complete high school with relatively low mathematics achievement scores are more likely to be unemployed or paid lower wages [2]. Similarly, Parsons and Bynner (2006) have run a large UK cohort study, finding that poor mathematics skills have a greater impact on life chances than do poor literacy skills [3].

When low achievement in mathematics is not due to low child intelligence or inadequate schooling, it is referred to with constructs such as “mathematical learning disabilities”, “mathematical learning difficulties” (MLD), and “developmental dyscalculia”. In this article, we will refer to this as MLD. The general terminological confusion is due to the fact that currently there is no clear generally accepted classification of developmental mathematical weaknesses [4]. Mathematical achievement tests are mostly paper-based and they assess different mathematical topics depending on the curriculum of each country. They often have served as screening measures to identify MLD students as those falling below determined cut-off scores; however, in different studies, these range from the second to the 46th percentile [5,6,7,8]. Of course this way of identifying children with MLD leads to heterogeneous groups of MLD children in which individuals can have different cognitive profiles [9].

Indeed, recent research suggests that students with MLD constitute a heterogeneous group and that trying to reduce atypical arithmetic development to one underlying core number deficit is too simplistic [10,11]. For instance, Fias, Menon and Szucs (2013) have highlighted how the development of number processing and mathematical problem solving is built on multiple neurocognitive components (such as working memory, approximate number system, etc.) and how impairment in any of these could compromise this learning [12]. Similarly, Skagerlund and Träff (2016) have shown that some MLD result from weaknesses with symbolic number processing, others from weaknesses with both symbolic and nonsymbolic number processing and that more general cognitive deficits can also be the cause of particular MLDs [13].

Given the significant economic and societal impact of low achievement in mathematics, the limitations of the curriculum-based achievement tests and the fact that mathematical achievement is not a uniform construct, there is a crucial need for a more comprehensive and widely shared test to both identify the children who are struggling with mathematics in general and also capture their specific sets of strengths and deficiencies. This sort of test should be able to assess children’s mathematical profiles, which can be used to tailor efficient intervention programs. In this paper, we present a tool developed to meet these goals. The Mathematical Profile Test (MathPro Test, MathPro Education, Athens, Greece) is a new computer-based battery that uses multiple measures and that proposes a broad array of math-related tasks covering a wide range of numerical abilities relevant to mathematical learning (http://mathpro.education/en/).

### 1.1. The Theoretical Framework of the MathPro Test

Since MLD is a heterogeneous condition, we assume that MLD could be due to impairments in the core number domain or to some general cognitive domains, particularly the visual-spatial, the memory and/or the reasoning domains [14].

### 1.2. Core Number Domain

Many studies have been conducted on number magnitude processing and these have been recently reviewed in two meta-analyses. First, Schneider, Beeres, Coban, Merz, Schmidt, Stricker and De Smedt (2017) found that mathematical performance was more strongly correlated with performance in symbolic (Arabic digits) magnitude comparison tasks than in nonsymbolic (dot) comparison tasks (respectively, *r* = 0.302 and *r* = 0.241) [15]. Second, Schwenk, Sasanguie, Jörg-Tobias, Kempe, Doebler and Holling (2017) found that children with MLD were significantly slower than controls in symbolic comparison tasks (Hedges’ *g* = 0.75) and, to a significantly lesser extent, on nonsymbolic (dot) tasks (Hedges’ *g* = 0.24) [16]. Another number magnitude processing that has been studied is the subitizing, i.e., the fast and accurate assessment of a small number of dots (range: one to four dots). Impairment in this process in children with MLD has been found in several studies (e.g., [17,18,19]). Finally, recent research has been considering another basic aspect of numbers: ordinal value. These studies have suggested that children with MLD are slower than typically achieving children in reciting the number sequence [20]; moreover, they have suggested that judging whether three numbers are presented in order or not is a very strong predictor of achievement in mathematics, even stronger than performance on number magnitude comparison tasks [21].

### 1.3. Memory Domain

Besides the core number processes described above, general cognitive processes are also important for appropriate numerical development. One of these is memory. Indeed, mathematics requires memorizing information and retrieving it from long-term memory. This is the case for the number words and their order in the count list or the products of certain arithmetical operations (i.e., arithmetical facts). Some studies have shown that children with MLD are slower than their peers in reciting the number words when counting [20] and several researchers have found that children with MLD have difficulty memorizing arithmetical facts (e.g., [22,23]). The difficulties in memorizing such facts may not be explained by globally weak long-term memory capacities [24,25,26] but by an exaggerated sensitivity to similarity interference in memory [25,26]. A difficulty in sequence memory has also been shown to correlate with poorer arithmetic fact processing in undergraduate students [27]. Many more studies have dealt with short-term memory capacities, finding that all working memory components (including inhibition, shifting and updating) are associated with mathematical performance, with the highest correlation being with verbal updating (see the meta-analysis [28]) and, among the different numerical domains, correlations with working memory (WM) are stronger for word-problem solving and calculations (see the meta-analysis [29]). Finally, children with MLD have poor verbal and visual-spatial working memory (WM) (see the meta-analysis [30]).

### 1.4. Visual-Spatial Domain

The visual-spatial domain is also related to mathematical achievement. Indeed, mathematics obviously requires visual-spatial processing for geometry but also for understanding other aspects, such as the place-value system of Arabic numbers or for solving vertical multidigit calculations. The use of diagrams reduces errors on word-problem solving because the use of visual images correlates with higher performance in word-problem solving [31]. Mathematical abilities also correlate with spatial skills; Rourke and his colleagues (and [32,33]) have argued that a particular type of MLD can be due to visual-spatial difficulties [34,35,36]. In support of this, Mammarella and colleagues have observed that children with impaired visual-spatial abilities, but intact verbal abilities, perform less well than typical-achieving peers in geometry [37], in written calculation and in number ordering [38]. Crollen and colleagues have gone further, showing that the number magnitude representation of these children is impaired [33,39]. Indeed, the number magnitude representation is supposed to be spatially oriented [40]. In these children, such a representation is less precise, and its left-right orientation is less strongly established. Yet, not all MLDs can be attributed to poor visual-spatial skills. For instance, Szucs, Devine, Soltesz, Nobes and Gabriel (2013) have not found any significant difference between MLD and control children on two visuospatial tasks (mental orientation and spatial symmetry); however, they have found poorer visuospatial short-term and working memory in these children [41]. A well-known task used to measure the mapping of numbers onto space is number line estimation. The meta-analysis of Schneider et al. (2018) shows that precision on this task significantly correlates with broader mathematical competence, concluding that this task is a robust tool for diagnosing and predicting mathematical competence [42].

### 1.5. Reasoning Domain

Reasoning skills are obviously important to learning mathematics. For instance, Fuchs and colleagues (2005) have found that nonverbal reasoning measured in the beginning of grade 1 predicted mathematics word-problem solving performance at the end of the school year [43] and Nunes and colleagues (2007) have also shown that, in six-year-old children, the logical abilities predicted their mathematical achievement 16 months later [44]. A similar association between nonverbal reasoning and word-problem solving has been observed in grade 3 children [45]. Across a wide age range (6 to 21 years), Green and colleagues (2017) found that fluid reasoning is a significant predictor of mathematical outcomes 1.5 and 3 years later [46]; Deary and colleagues (2007) have found that fluid intelligence tested at age 11 predicted about 60% of the variance of math performance at age 16 [47]. Moreover, children with MLD have been found to develop weaker reasoning skills than typically achieving children [48]. Finally, training children in logical reasoning has been shown to lead to higher progress in mathematics than within a control group who did not receive this training [44]; this supports the idea of a causal link between logical reasoning and mathematical learning.

### 1.6. Selection of the MathPro Test Tasks

Taking into account the above literature, Karagiannakis and colleagues (2017) proposed a four-pronged classification model according to which MLD could be characterized by problems arising in core number processes, deficits in memory, visuospatial processes or reasoning processes [49]. Indeed, weaknesses in students with MLD may reside in a single one of these domains or within any combination of them. The domains are not considered, a priori, to be hierarchical in any way. These hypotheses were confirmed through analyses of data collected by administrating an experimental computerized battery based on this four-pronged classification model of the basic mathematical skills of Greek students of ages 10–12 years (fifth and sixth grade). The battery included 13 numerical subtests, with the a priori assumption that certain sets of tasks tap more strongly into a particular domain of the four-pronged model. The battery was administered to a sample of 165 typical students. Exploratory and confirmatory factor analyses were run, leading to the extraction of four factors: (1) a *core number* skills factor that included the dots magnitude comparison, the subitizing-enumeration and the number magnitude comparison subtests; (2) a *visual-spatial* factor that included the number lines 0–100 task; (3) a *memory* factor that included the multiplication facts retrieval and the addition facts retrieval tasks and (4) a *reasoning* factor that included the mental calculations equations, the word problems, the number lines 0–1000, the math terms and the calculations principles tasks.

Although it was a very promising battery for collecting data for research purposes, it had a number of limitations. First, only one task was loaded on the visual-spatial factor. Second, response times were recorded but there was no control of the user’s global processing speed, such as the speed in using the computer mouse or the keyboard. Third, there were other limitations regarding the method used on some subtests. For instance, in the dots magnitude comparison tasks, the pairs of dots remained on the computer screen until students responded, allowing the students to potentially count the dots, while the task was supposed to assess estimation skills. Fourth, in the multiplication and addition facts retrieval subtests, students had to select the correct answer from two choices; this raised the possibility of a correct answer as a result of a random choice. Finally, the generalizability of the results of the study was limited due to the restricted age range of the participants.

The MathPro Test was developed to address such limitations and present a more advanced tool to assess the four domains of numerical skills. More precisely, compared to the previous battery, the following changes were made: (1) two new tasks were developed to assess visual-spatial mathematical skills, the squares and the building blocks tasks; (2) a screen keyboard-use task was developed to assess the motor reaction time in using both the mouse and the screen keyboard to respond; (3) in the dots magnitude comparison task, the dot arrays were shown for a fixed period of 840 milliseconds (ms) and (4) production rather than verification was used in the facts retrieval tasks (multiplication and addition facts retrieval), with the student composing the answers on the screen keyboard. Furthermore, the word-problem task was modified: students only had to indicate the arithmetical operation needed to solve the problem rather than solve it. This modification was introduced so that pure reasoning skills and not computational skills would be assessed. Finally, three new tasks were introduced: the next number and the previous number tasks tapping into number ordinality, which has proven to be a highly significant predictor of math skills [21] and a numbers dictation task assessing number transcoding from spoken number word to the Arabic format to assess more deeply the mastery of symbolic numbers.

The subtests of the MathPro Test are grouped based on the a priori assumption that they tap more strongly into one of the aforementioned domains even though most of them actually tap into several of these domains. To be more specific, the subtests included in the *core number* domain attempt to elicit both the approximate (i.e., dots comparison) and exact (Arabic digit comparison) numerical magnitude processing skills as well as subitizing skills. The *memory* domain includes tasks that involve remembering number words (number dictation) or accessing memorized sequences (such as the counting sequence, knowing which number comes before or after another) and arithmetical facts (single-digit additions and multiplications). The *reasoning* domain includes tasks that involve using the place value system (multidigit number comparison), word-problem solving (decision making in particular), using basic calculation principles (deductive reasoning to implement) and identifying numerical patterns (through inductive reasoning). The ability to solve multidigit mental calculations is also related to this domain although, in addition, it requires memory (because some arithmetical facts need to be retrieved and intermediate solutions stored in short-term memory). The *visual-spatial* domain includes tasks that require analyzing 2D and 3D geometrical figures and also number line estimation tasks, because it requires spatial skills [50] and visuomotor integration [51]. We note that the number line tasks are also sensitive to intelligence [52] and could therefore also be somewhat related to the reasoning domain.

### 1.7. The Aim of This Study

The current study extends the study of Karagiannakis et al. (2017) by addressing the limitations of the previously presented experimental battery and by testing a much greater population, from grade 1 to grade 6, from another country (here, Belgian Dutch-speaking children) [49]. As underlined by Reigosa-Crespo et al. (2012), most of the studies have focused on an age when children are first being introduced to formal mathematics [53]. As a consequence, there is a lack of research that examines the acquisition and the development of more complex and sophisticated arithmetic skills. Taking into account children from grades 1 to 6 will contribute to this understanding.

The main goal of the MathPro Test is to gain insight into students’ mathematical skills pertaining to each of the four domains and potentially provide information to educators (regular classroom teachers, after-school coaches and clinicians who offer remedial intervention) about the strength and weakness of the student they are working with.

The specific aim of this paper is to present the preliminary evaluation both of the sensitivity and the psychometric characteristics of the individual measures of the MathPro Test across 1–6 grades. This will help us to determine how it can be utilized not only in research but for educational purposes as well. To that aim, for each task, we measured the internal consistency and the variability of children’s performance according to their school grade and the level of difficulty of the items. To obtain a first idea of the validity of the subtests, a global math performance test was also administered to each student and correlations between each task of the battery and this global measure were calculated. Finally, based on this global math performance test, we identified children with difficulties in mathematics (MD) and compared their performance on the battery with those of typically achieving (TA) children.

## 2. Materials and Methods

The participants in this study were 622 students in grades 1–6 from seven elementary schools in Flanders, Belgium. They all spoke Dutch. Parental consent was obtained for all students. Participants came from a variety of socio-economic backgrounds (which, unfortunately, were not made available for the study). Table 1 shows the descriptive statistics of these students according to their grades and gender. The numbers of boys and girls did not differ across grades; $x^{2}$ (5, N = 622) = 2.91 and *p* = 0.713.

The research was conducted in agreement with the Helsinki ethics declaration [54]. Schools and parents received information about the research purpose and procedure. They all gave written informed consent. Students were tested in groups in computer classrooms of their schools. They were informed that they could interrupt the test any time they wanted without any penalties. The data collection took place during the spring period within a period of four weeks. Data were totally anonymous; they were saved in files using an identification code with no personal information regarding the children associated with it.

### 2.1. MathPro Subtests

The MathPro Test includes 18 subtests measuring mathematical skills in the following domains: *core number, memory, reasoning* and *visual-spatial* plus one subtest measuring the processing speed (screen keyboard use). The subtests are classified into the aforementioned four domains introduced above, assuming that each subtest elicits the cognitive abilities of primarily one domain (see Table 2). Each of these subtests will be described in detail in the order in which they are presented in the battery administration.

For each subtest, instructions are provided through animations simultaneously with the audio. Text instructions are also available by clicking on a particular button. Instructions contain an explicit presentation of the task of each subtest as well as tips (for example, *“Pay attention: the size of the dots does not matter. They can be small or big”* or *“There could be building blocks hidden behind other building blocks”*). Instructions can be repeated several times on demand. After the instruction video, three practice trials follow. Each practice trial is accompanied by corrective feedback provided both in visual and auditory modes. The incorrect responses at this stage are accompanied by extra instructions. Between the practice level and the real task, there is an audio reminder when reaction time is also calculated: *“Remember: in this task, time counts. Answer as quickly as you can, but without making mistakes!”* In each subtest, trials start with a fixation sign at the center of the computer screen, shown for 250 ms. The stimuli appear after a 1000 ms pause. For subtests where the child should be fast and the reaction time is measured, a clock drawing remains on the right top corner of the screen to remind the child that the response time is measured. Response time is measured with millisecond precision.

### 2.2. MathPro Subtests

#### 2.2.1. Dots Magnitude Comparison

Two collections of dots are simultaneously presented on the computer screen and students are asked to select the one that contains more dots (*“Which picture has more dots?”*) by clicking on it with the computer mouse (Figure 1). Sets of black dots were created based on Gebuis and Reynvoet’s work (2011), and the MATLAB code publicly provided by the authors [55]. The pairs are ordered by increasing ratio difficulty, that is, 2:3, 3:4, 4:5, 5:6 and 6:7. There are three comparison pairs, with each ratio appearing twice, once in ascending and once in descending order. In half of the trials in each ratio, numbers are congruent with physical cues (convex hull, size of the dots or density), whereas the other half are incongruent. In total, there are thus 30 items. The pair of stimuli appears for 840 ms to prevent the possibility of counting. Accuracy is computed.

#### 2.2.2. Single and Multidigit Number Magnitude Comparison

Two Arabic numbers are presented; students are asked to decide as quickly as possible which is the larger number (*“Which is larger?”*) by clicking on it with the computer mouse. The Arabic numbers are written in a dark color on a white circle background. There are 36 comparison pairs in total: 24 single-digit number pairs (subtest 2) and 12 multidigit number pairs (subtest 3). Single-digit number pairs are made up of six close number pairs (distance 1, e.g., 5–6) and six distant number pairs (distance of 4–5, e.g., 9–4), appearing once in ascending and once in descending order. Then, three two-digit pairs (e.g., 27–31) with a decade-unit incompatibility (i.e., the larger number has the smaller digit in the unit position) are presented in the two orders. Then three three-digit pairs (e.g., 109–170), three four-digit pairs (e.g., 1100–1080) and finally three decimal numbers pairs (e.g., 0.3–0.20) that are sensitive to the whole number bias [56] are presented. The response side is counterbalanced. The single-digit number pairs are intended to assess the speed of accessing Arabic number magnitude, whereas the last 12 items assess the student’s understanding of the positional value of the 10-based numerical system. The pair of stimuli appears and remains until the students provides a response. Both accuracy and reaction time are computed.

#### 2.2.3. Screen Keyboard Use

Students are presented on the computer screen with a single Arabic number and are asked to indicate it as quickly as possible (*“Which number is this?”*) by clicking with the mouse on the screen keyboard that appears on the screen (Figure 2). The keyboard includes a backspace button to allow the student to change his/her response and an input response button that the child needs to press to validate his/her answer. The reaction time is measured from the time stimuli are presented until the child clicks the validation (tick) button to validate the answer. In this subtest, 10 single-digit numbers are presented and each of them remains until the student presses the validation button to validate his answer and proceed to the next trial. This subtest is intended to measure the motor reaction time of using both the mouse and the screen keyboard in order to control this in the other numerical tasks that use the keyboard for the child’s response production.

#### 2.2.4. Numbers Dictation

Number words are presented through computer speakers and students are asked to click on the screen keyboard (Figure 3) the Arabic digits corresponding to the heard number (*“Which number did you hear?”*). Stimuli are 30 items, varying from single-digit to five-digit numbers (six items for each digit length) presented in order of increasing number of digits. As transcoding numbers with zeros, especially with internal zeros, are more challenging for students [57], all three-, four- and five-digit numbers contained one, two and three zeroes, respectively, either in the middle or/and the last position (e.g., 601, 2050 and 70,020). The task stops after three consecutive errors. Both accuracy and reaction time (starting from the beginning of the sound file in this case) are computed. The next trial starts when the child has pressed the tick button.

#### 2.2.5. Next Number

Number words are presented through the computer’s speakers and students are asked to indicate as quickly as possible the following number (*“Which number comes just after?”*) by clicking on the corresponding digits on the screen keyboard. Eighteen items are presented: six single-digit, six two-digit and six three-digit numbers, presented in order of increasing number of digits. As jumping from one decade to the other or from one hundred to the other can be challenging, numbers ending with nine were selected more often than those ending with any other specific digit (this was the case for half of the two-digit and half of the three-digit numbers). The task stops after three consecutive errors. Both accuracy and reaction time are computed.

#### 2.2.6. Previous Number

This task is the same as “The next number” except that here students are asked to produce the preceding number: *“Which number comes just before?”*. There are 18 items in total, varying from single-digit to three-digit numbers. As going from one decade to the previous one or from a hundred to the previous one can be challenging, numbers ending with zero were selected more often than those ending with any other specific digit (this was the case for eight items). Items are presented in order of increasing number of digits. Both accuracy and reaction time are computed. The task stops after three consecutive errors.

#### 2.2.7. Subitizing

Small arrays of black dots are very briefly presented on the computer screen and students are asked to indicate the corresponding numerosity (*“How many dots were there”*) by clicking on the screen keyboard. Random configurations of two to six dots are presented (four different configurations for each number) in a fixed order (with never the same number in two consecutive trials and no three consecutive numbers appearing consecutively). Stimuli appear for 300 ms and are then covered by a black and white squared grid to avoid counting on an afterimage (Figure 4). The child enters his/her response and then presses the tick button. Accuracy is computed.

#### 2.2.8. Enumeration

Students are presented with arrays of seven to 13 black dots (arranged randomly) and are asked to count them and indicate the cardinal (*“How many dots are there?”*) by clicking on the screen keyboard. A total of 14 items (seven set sizes and two repetitions) are presented in a random fixed order (never the same set size consecutively and never three consecutive numbers appearing in a row). The stimulus appears and remains until the child’s response. Both accuracy and reaction time are computed.

#### 2.2.9. Addition Facts Retrieval

Single digit additions with operands from two to nine and a sum equal to or below 10 are presented in the Arabic code and students are asked to indicate as quickly as possible the correct sum (*“What is?”*) by clicking on the screen keyboard (Figure 5). There are 12 stimuli. The stimuli remain until the pupil presses the tick button to validate his/her answer and proceed to the next trial. Both accuracy and reaction time are computed.

#### 2.2.10. Multiplication Facts Retrieval

This subtest follows exactly the same procedure as the addition facts retrieval subtest except that, here, 14 single-digit multiplications are presented. The factors range from two to nine but at least one of them is equal to or below five.

#### 2.2.11. Mental Calculations

Addition, subtraction, multiplication and division operations with numbers up to three digits long are presented to students, who are asked to indicate, as fast as they can, the correct answer. There are 24 stimuli in total. The six trials of each operation are ordered in terms of increasing difficulty. A warning video precedes all new operations in order to draw the child’s attention to such a change; this was designed to decrease possible operation shifting errors. The same procedure as for addition facts retrieval subtest is used here. Both accuracy and reaction time are computed.

#### 2.2.12. Number Lines 0–100

Number lines 0–100 are presented on the computer screen. An Arabic target number from 0 to 100 appears above the center of the number line (Figure 6). Students are asked to mark the correct location of that target number on the line (*“Where would you put this number?”*) by selecting a position with the mouse. Students can change the position of the number by dragging the point left or right. To validate their choice, they need to click the tick button. The 22 target numbers presented are the same as those used by Siegler and Opfer (2003) or Siegler and Booth (2004), with numbers in the lower range being over-represented to allow discrimination between logarithmic and linear functions [58,59]. The stimuli are ordered randomly. Each stimulus remains on the screen until the child validates a response. The number line coordinates for each response are recorded based on a pixel count along the length of the line. Accuracy of estimates is defined here as the absolute difference between the student’s estimation of the target number and its correct position. The percentage of absolute errors (PAE) is used as the measure.

#### 2.2.13. Number Lines 0–1000

The same task as in number lines 0–1000 is presented using a line marked with 0 at the left end and 1000 at the right end. Twenty-two stimuli are presented [60].

#### 2.2.14. Squares

Stimuli are geometrical shapes built on a white grid square background by joining a combination of blue squares, blue half-square triangles and blue quarter-square triangles. Students are asked to indicate the total number of whole squares that can be made from each configuration (*“How many whole blue squares can be made?”*) by clicking on screen keyboard. Stimuli are differentiated in terms of both the combination of the parts composing the configuration (squares, half-square triangles and quarter-square triangles) and the ratio of visible sides to the total number of sides of the squares (the rations are: 1, 3:4, 2:3, 4:7, 2:5, 1:5 and 0). The first three geometrical shapes contain only whole squares (Figure 7a), while the others contain a combination of whole squares and half-square triangles (Figure 7b) and then quarter-square triangles are also included (Figure 7c). Each stimulus appears and remains until the student responds and presses the tick button. Accuracy is computed.

#### 2.2.15. Building Blocks

Stimuli are 3D structures made of cube building blocks (and represented in 2D on the screen). Students are asked to indicate the number of cubes that each 3D structure contains in total (*“How many building blocks are there?”*) by clicking on the screen keyboard (Figure 8a). The structures are made of five to 10 blocks and some of these (up to three) can be hidden behind the others (Figure 8b). A stimulus appears and remains until the student responds and presses the tick button. Accuracy is computed.

#### 2.2.16. Word Problems

Word problems are presented to the students, who are asked to specify the calculation they would implement to solve the problem: *“How would you solve this problem?*” Students have to indicate on the screen keyboard (Figure 9) the way each problem should be solved (for example, “37 + 25” and not the final solution “62”). One-step word problems involving addition, subtraction, multiplication and division are presented. In order to reduce the possible impact of reading difficulties, the problems are one to three sentences long and they appear written on the computer screen and can also be read aloud (with the possibility to ask for a repetition of the problem by clicking on the speaker icon). Eighteen problems were selected. Thirteen of these are addition-subtraction problems [61], among which are five comparison problems, two combination problems and one equalization problem. The other five trials consist of multiplication-division problems [62]. A problem appears and remains until the child responds and presses the tick button. Stimuli are ordered in increasing difficulty based on a pilot study. The task stops after three consecutive incorrect responses. Accuracy is computed.

#### 2.2.17. Calculation Principles

A pair of related multidigit operations, one with the correct answer, the other without the answer given, are presented to the student (Figure 10). Students are asked to indicate the solution to the second problem without calculating it but with reference to the first problem: *“What is the second if you know the first?”* The following principles are evaluated: commutativity of addition (e.g., “if 37 + 48 = 85, what is 48 + 37 = ”), addition/subtraction inversion (e.g., “if 68 + 25 = 93, what is 93 − 68 = ”), sum minus one (e.g., “if 35 + 47 = 82, what is 35 + 48 = ”), difference plus or minus one (e.g., “if 60 − 27 = 33, what is 61 − 27 = ”), 10a + 10b (e.g., “if 57 + 86 = 143, what is 570 + 860 = ?”), multiplication as repeated additions (e.g., “if 78 × 4 = 390, what is 78 + 78 + 78 + 78 = ”), commutativity of multiplication (“if 47 × 18 = 799, what is 18 × 47 = ”), multiplication/division inversion (e.g., “if 518 ÷ 37 = 13, what is 37 × 13 = ”), (a + 1) × b (e.g., “if 146 × 7 = 1022, what is 147 × 7 = ”), and 10a × 10b (e.g., “if 36 × 7 = 252, what is 360 × 70 = ”). Each of the 15 stimuli appears and remains until the student responds and presses the tick button. Stimuli are ordered in increasing difficulty according to the data of a pilot study. The task stops after three consecutive incorrect responses. Accuracy is computed.

#### 2.2.18. Numerical Patterns

A series of numbers, with one missing, is presented horizontally in the center of the computer screen and the child is asked to find the missing one (*“Which number is missing?”*) and type it on the screen keyboard. The numerical patterns include arithmetical series (e.g., 1, 3, 5, 7, _), geometrical series (e.g., 80, 40, 20, _) and more complex series (e.g., 2, 4, 5, 10, 11, 22, _). Each of the 18 stimuli appears and remains until the pupil responds and presses the tick button. Stimuli are ordered in terms of increasing difficulty based on the data of a pilot study. The task stops after three consecutive incorrect responses. Accuracy is computed.

The younger students (grades 1, 2 and 3) did not take the whole MathPro Test following the curriculum in math in Flanders. Some subtests were not presented to the younger students and, in some other cases, only the first easier items were presented (see Table 3).

### 2.3. MathPro Test Advantages

The MathPro Test is an online computer-based tool, which presents several advantages. First, the computer-based administration allows clear measurement of reaction times as well as the designing of tasks in which the stimuli need to appear for a very short time. Indeed, tasks such as dots magnitude comparison or subitizing could not be implemented reliably without the support of a computer, and they are necessary for assessing specific numerical skills. Second, the MathPro Test does not necessarily require an experimenter to assist the child as all assignments and responses are given exclusively through interaction with the computer; therefore, it provides unbiased administration of the test to all students and no one needs to be trained to give the test. Third, the online administration provides an easy-going procedure both for one-to-one and collective administration regardless of where students live. All recorded data are securely saved in real time on the test’s server and they can be extracted to an Excel file at any time. Fourth, the MathPro Test produces a complete profile of the child within only 45 to 60 min. The report is extracted automatically immediately after completion of the test; this prevents any human calculation errors. The report includes students’ scores with respect to both reaction time and accuracy compared with their peers’ norms; the report also provides an error analysis. Consequently, the administrators do not have to follow the time-consuming procedure of calculating the standard scores. Fifth, the MathPro subtests are adapted to each grade of primary education by establishing in advance different ending points or through stopping criteria (for example, some subtests are programmed to stop after three consecutive errors when their items have been ordered in terms of increasing difficulty) on certain subtests. Finally, the system is developed in such a way that the battery can be translated almost automatically into any language (the MathPro test is currently available in Dutch, English, French, Greek, Italian and Maltese). This facilitates the carrying out of international comparison studies.

### 2.4. Standardized Test in Mathematics

In general, mathematics achievement was assessed using a curriculum-based standardized achievement test for mathematics from the Flemish Student Monitoring System [63]. This test is untimed and includes 60 items covering number knowledge, understanding of operations, (simple) arithmetic, word-problem solving, measurement and geometry. The items of the test differ according to the school grades as it is based on the mathematical content the child has learned in math in the months preceding the test. The score on this achievement test is the number of correctly solved items (maximum = 60). Norms were available for all grades of primary school.

### 2.5. Statistical Analysis

Statistical analyses were run on the IBM SPSS 25 [64]. First, internal consistency of each task was assessed using Cronbach’s alpha. When the items presented differed between the grades, different Cronbach alphas were calculated. Second, in order to examine the grade effect, ANOVAS (or ANCOVA for reaction times (RTs) measures controlling for general motor speed) were used on each of the measures provided by the battery (considered as dependent variables), with the student’s school grade as between students’ factors. Third, to further measure the sensitivity of the battery, repeated measures ANOVAs were run separately for each measure of the battery (dependent variable) with the task’s difficulty level as a within-subject factor. Huynh–Feldt and Greenhouse–Geisser corrections were used when the sphericity was violated. We expected performance to increase with school grade and to decrease with the increasing level of task difficulty. Fourth, to measure concurrent validity, zero-order correlations were measured between measures from the battery and the standardized test in mathematics (STM). Finally, as a first attempt to assess the discriminative power of the battery, we examined the extent to which performance on the difference tasks of the battery could identify the students with difficulties in mathematics (or MD). To this end, after we split the sample into MD and control students using as selection criterion their score on the STM, one MANOVAs for accuracy and one MANCOVA (controlling for general motor speed) for reaction time were conducted separately for first second and third graders since the subtests or/and the items within the subtests were different according to the grades. One MANOVA for the accuracy and one MANCOVA for the RTs were run for the fourth to sixth graders since they took the whole version of the test.

## 3. Results

### 3.1. Descriptive Analyses

Table 4 shows students’ performance per grade in terms of accuracy (AC: number of correct responses divided by the number of items) for all subtests of the MathPro Test and reaction time (RT) for the subtests where the response time was measured. The percentage of absolute errors (PAE) was used for both the number line tasks. We analyzed students’ reaction times for the items of each subtest when accuracy was above 0.85 (otherwise, students who only succeeded on the easier items might have had shorter response times than those who went on and succeeded also on the more difficult items). For this reason, we excluded from the analyses of the reaction times the decimal numbers items of the multidigit numbers comparison task, the five-digit numbers of the numbers dictation subtest and the three-digit numbers of the next and previous numbers subtests. We note that in all the analyses relating to a child’s RTs in producing an answer using both the computer mouse and the screen keyboard, RT measured in the screen keyboard use task was introduced as a covariate variable.

### 3.2. Internal Consistency

Cronbach alphas were calculated for all the dependent measures. When the same task was presented to all the grades, a single Cronbach alpha was calculated. When the items presented differed between the grades, different Cronbach alphas were calculated, and their range is provided (see Table 5). Good or excellent internal consistency (α > 0.8) was obtained for 19 measures, acceptable (0.6 < α < 0.7 for four measures, poor (0.5 < α < 0.6) for four measures and unacceptable (α < 0.5) for one measure [65].

### 3.3. Grade and Difficulty Effects

For all the tasks, repeated measures ANOVAs (or ANCOVAs for RTs measures) were run to assess the effects of school grade and the level of difficulty of the task (entered as a within-subject factor). Huynh–Feldt and Greenhouse–Geisser corrections were used when the sphericity was violated. The difficulty levels were the following: five levels in the dots comparison (ratios: 2:3, 3:4, 4:5, 5:6 and 6:7), two levels in the single-digit comparison both for AC and RT (close and distant pairs), four levels in the AC for multidigit comparison (two-digit, three-digit, four-digit and decimal numbers) but three levels only for the RT (two-digit, three-digit and four-digit as AC < 0.85 for the decimal numbers), five levels of difficulty for AC in the number dictation (one-, two-, three-, four- and five-digit numbers) and four levels for the RT (five-digit numbers excluded because they did not meet the >0.85 accuracy), three levels of difficulty for AC in the next and previous numbers (one-, two- and three-digit numbers) but two levels for RT (three-digit numbers excluded due to the accuracy criterion noncompliance), five levels of difficulty (two, three, four, five and six dots) for the subitizing, two levels for the enumeration (equal to or less than 10 dots and more than 10 dots), two levels for at addition facts retrieval (sum equal to or less than eight and more than eight), two levels for the multiplication facts retrieval (factors’ sum equal to or less than 10 and more than 10), three levels for the mental calculations (2 × 4 operations for each level based on the numerical size of the operands), two levels for the squares, depending of the ratios of visible sides to total sides of the squares (equal to or less than 4:7 and more than 4:7), two levels for the building blocks (none or one hidden block and two to three hidden blocks) and, finally, three levels of difficulty for the word problems and for the calculation principles and two for the numerical patterns defined based on a pilot study. For the number line tasks, the 0–100 number line task was considered as easier than the 0–1000 number line task. Accordingly, a repeated measure ANOVA on the PAE was calculated measuring the effects of school grade (between student factor) and the effect of the type of number line task (within student effect) as an index of difficulty level.

A significant grade effect (all *p* < 0.001) was found in all the tasks and for all the measures (AC and RT), indicating that, as expected, performance increased with the student’s school grade (see Table 6 for the ANOVAs or ANCOVAs results). The difficulty level was significant for all accuracy measures but the addition facts retrieval. For RTs, the difficulty effect was significant for the single and multidigit comparisons and the number dictation but not for the other tasks.

On the dots comparison task, AC decreased significantly as the ratio approached 1 (ratio 2:3 (0.68 ± 0.16), ratio 3:4 (0.62 ± 0.17), ratio 5:6 (0.60 ± 0.19) and ratio 6:7 (0.54 ± 0.2)), except for the ratio 4:5 (0.76 ± 0.17). For single-digit comparison (Figure 11a), number pairs with a large distance were processed significantly more accurately and faster than close pairs.

For multidigit number comparison, AC (for the grade 4–6 students) was similar for three-digit (0.96 ± 0.12) and four-digit (0.95 ± 0.13) number pairs and was slightly lower for two-digit numbers (0.92 ± 0.16) and much lower for the decimals (0.60 ± 0.40). The surprising difference between the three- and four-digit numbers on the one hand and the two-digit numbers on the other is probably due to the fact that all two-digit number pairs were made of decade-unit incompatible pairs. Regarding RTs, the four-digit number pairs took students significantly more time (3131 ± 541) than the two-digit (2859 ± 655) and the three-digit numbers (2842 ± 412), which were processed with similar speed. For numbers dictation, RTs increased significantly with the number of digits, whereas AC (Figure 11b) decreased significantly for numbers with more digits. In the next number task, AC decreased significantly with the increasing of the number of digits (one-digit (0.94 ± 0.14), two-digit (0.91 ± 0.21) and three-digit (0.86 ± 0.26)). This was also the case for the previous number both for the AC (one-digit (0.97 ± 0.11), two-digit (0.94 ± 0.17) and three-digit numbers (0.87 ± 0.23)) and the RTs (one-digit (2527 ± 413) and two-digit numbers (3627 ± 799)). On the subitizing task, AC significantly decreased (Figure 11c) when the number of dots increased. Students were significantly more accurate on enumeration for the smaller collections of dots. In multiplication facts, students were significantly more accurate on the small products (0.91 ± 0.18) than the large (0.88 ± 0.20) ones. Students’ AC in mental calculations decreased significantly with the increased difficulty (easy (0.85 ± 0.19), medium (0.82 ± 0.22) and difficult (0.66 ± 0.28)). For the number line tasks, the PAE was significantly lower for the 0–100 than for the 0–1000 scale (Figure 11f). On the squares task, students performed significantly better when the ratio of the visible sides of the squares was large (large (0.90 ± 0.18) and small (0.48 ± 0.27)). On the blocks task, AC (Figure 11d) was significantly higher when the number of hidden blocks decreased. For word problems, students’ accuracy decreased significantly along with the increasing difficulty level (easy (0.73 ± 0.29), medium (0.50 ± 0.41) and difficult (0.37 ± 0.39)). This was also the case for the calculation principles (Figure 11e). Separate analyses of the difficulty effect were also run for the earlier grades separately (see Appendix A).

Furthermore, the grade × difficulty level interaction was significant for 16 of the measures. For the sake of brevity, all these interactions will not be described in detail. Five of them are illustrated in Figure 11. Globally, these interactions indicate that sensitivity to the difficulty of the tasks is usually larger for younger students than for older ones. Yet, it is important to note that, for each of these measures (but two) that showed a significant grade × difficulty level interaction, the difficulty effect remained significant when only sixth graders were considered. This indicates that tasks remained sensitive even for the older students. The two exceptions are the next and previous number tasks, where the difficulty effect was no more significant in grade 6 for AC, but it was for RT.

### 3.4. Correlations with the Standardized Math Test

Correlations were calculated between the score of the standardized math test (SMT) and each subtest from the MathPro Test separately for each grade (Table 7). Significant correlations were obtained with the dots comparison task for middle grades but not for grade 1 or grades 5 and 6. For the other basic numerical tasks (single-digit numbers comparison, next number, previous number, subitizing and enumeration), significant correlations were only observed in grades 1 and 2. The correlations for the building blocks were significant in all the grades but grades 2 and 5 and for multidigit numbers comparison, numbers dictation, calculation principles and the addition and multiplication facts retrieval correlations were significant up to grade 5. Finally, significant correlations were found in all the grades for mental calculations, number lines, squares, word problems and numerical patterns. To examine the correlation between the MathPro Test and the SMT in total, a global score was calculated by transforming the raw scores of the subtests to standardized z-scores separately for each grade. Their sum was used as the MathPro index score (note that PAE z-scores where multiplied by −1 as a high score in both cases actually means a low performance). The correlation between the MathPro index score and the SMT was very strong in grades 1–5, ranging from 0.57 to 0.64 and less strong (0.47) in grade 6.

### 3.5. Comparison of MD Versus Typically Achieving Students

Finally, to examine the extent to which performance on the different tasks of the battery could identify students with difficulties in mathematics (or MD) with respect to the others, we considered students whose achievement in the SMT was equal to or below the 15th percentile as MD students (n = 16, *n* = 20, *n* = 35, *n* = 19 and *n* = 22, respectively, for grades 2, 3, 4, 5 and 6) and compared them to those who performed equally to or above the 30th percentile, who we considered as typically achieving (TA) students (*n* = 75, *n* = 73, *n* = 70, *n* = 60 and *n* = 82).

For grade 1, the 15th cut-off led to a very small MD group (*n* = 8). Accordingly, we selected a less stringent cut-off of the 30th percentile (*n* = 20 for MD students), keeping the same criterion for determining the TAs (*n* = 83). This resulted in a group of 132 MD students and 443 TA students. To compare their performances on the MathPro Test, two MANOVAs (one for accuracy and one for reaction time) were conducted separately for first, second and third graders since the subtests or/and the items within the subtests were different according to the grades. Since students in grades 4–6 took the whole version of the test, two MANOVAs were run, one on the speed measures, the other on accuracy measures for these three higher grades all together.

The MANOVA run on accuracy measures for grade 1 showed a significant global group effect (F (11, 90) = 4.301, *p* < 0.001 and η^2^ = 0.40). Indeed, MD students performed significantly less accurately than TAs in numbers dictation, subitizing, enumeration, addition facts retrieval, number lines 0–100, building blocks and numerical patterns. They were also slower in the single-digit numbers comparison, i.e., the sole task where their speed was measured (Table 8).

For grade 2, the MANOVA run on accuracy measures showed a significant global group effect (F (14, 89) = 2.167, *p* = 0.015 and η^2^ = 0.30). MD students performed significantly lower than TAs in single digit numbers comparison, multidigit numbers comparison, numbers dictation, previous numbers, subitizing, mental calculations, number lines 0–100, word problems and numerical patterns (Table 8). The MANCOVA run for the reaction time controlling for the general speed showed a significant global group effect (F(3, 84) = 4.532, *p* = 0.005 and η^2^ = 0.14). The MD students were significantly slower than TAs in addition facts retrieval.

For grade 3, the MANOVA run on accuracy measures showed a significant global group effect (F (18, 83) = 5.115, *p* < 0.001 and η^2^ = 0.56). MD students significantly underperformed on multidigit numbers comparison, numbers dictation, mental calculations, number lines 0–100 and 0–1000, squares, word problems and calculation principles (Table 8). The MANCOVA run on speed measures controlling for the general motor speed failed to show a significant global effect of the group (F(8, 82) = 1.896 and *p* = 0.064) as MD students were significantly slower than TAs in multidigit numbers comparison only (Table 8).

Finally, the MANOVA run on accuracy measures for grades 4–6 showed a significant global effect of the group (F (18, 246) = 7.221, *p* < 0.001 and η^2^ = 0.34) and the grade (F (36, 494) = 3.890, *p* < 0.001 and η^2^ = 0.22) but not of the group × grade interaction (F (36, 494) = 1.378 and *p* = 0.079). MD students performed significantly more poorly than TAs in dots comparison, multidigit numbers comparison, numbers dictation, multiplication facts retrieval, mental calculations, number lines 0–100 and 0–1000, squares, building blocks, word problems, calculation principles and numerical patterns (Table 9). The MANCOVA run on speed measures controlling for the general motor speed showed a significant global effect of the group (F (8, 263) = 3.451, *p* = 0.001 and η^2^ = 0.10), the Grade (F (16, 528) = 4.596, *p* < 0.001 and η^2^ = 0.12) but not of the group × grade interaction (F (16, 528) = 0.987 and *p* = 0.469). Indeed, MD students were significantly slower than TA students in multidigit numbers comparison, enumeration, addition and multiplication facts (Table 9).

## 4. Discussion

This research was conducted in response to the need within the domain of numerical cognition for an assessment tool that is theoretically driven, covering a wide range of key numerical processes, while being administrable in a reasonable amount of time. Indeed, such an assessment tool is fundamental for early identification of the learning profiles of students who are struggling in math.

In this paper, we presented a new battery, the MathPro Test, together with the preliminary examination of its sensitivity and psychometric characteristics of its individual measures. This battery was designed based on the pilot battery developed by Karagiannakis and colleagues (2017) [49]. The main limitations of the previous pilot battery were addressed and overcome in the current study, presenting a new battery able to assess the four domains of numerical skills (core number, memory, visual-spatial and reasoning) of students from grades 1 to 6. Some of the numerical tasks of the battery were similar to those included in previously published assessment tools. In particular, it included a symbolic number comparison task, as in the *Symbolic Magnitude Processing Test* [66] or in the *Numeracy Screener* [67]—a dots comparison task (as in the *Numeracy Screener* or in the *Panamath* [68]) and addition and multiplication facts retrieval tasks (as in the *Dyscalculia Screener*; [69]). However, it also included many other mathematical tasks, thereby allowing overcoming the achievements of recent tools that have been developed. We now discuss the sensitivity and the psychometric characteristics of the individual measures of the MathPro Test across grades 1–6.

Significant grade effects were obtained for the speed and accuracy measures in all the tasks of the battery, revealing a clear increase in students’ performance through the school years. Furthermore, on each of the tasks, except for addition facts retrieval, performance decreased with the increasing difficulty of the items. On most of the subtests, there was an interaction between grade and items difficulty. Indeed, as one could expect, older students have addressed more mathematical contents and they should be able to deal with both easy and more difficult items. Accordingly, sensitivity to item difficulty manipulation decreased as older grades were considered. The difficulty effect was indeed smaller than in lower grades; however, it remained significant for all tasks (except for the next and previous number tasks), when only sixth graders were considered. This suggests that the tasks remained sensitive up to grade 6.

The validity of the MathPro Test was also supported by the significantly strong correlations between the global MathPro index and the global math performance test used in the school, i.e., the STM. Although these results are very promising, at this point, it is interesting to consider each of the subtests and examine both their correlations with the STM and their ability to differentiate between MD and typically achieving (TA) students. As reported in Table 10, it appears that some subtests of the battery are more strongly related to the STM scores and lead to a greater difference between MD and TA students. These results will be discussed, considering each domain one after the other.

### 4.1. The Core Number Domain Measures

The subtests included in the *core number* domain attempted to elicit both the approximate (dots comparison task) and exact numerical magnitude processing skills (subitizing task and single-digit numbers comparison task). The dots comparison (AC) and the single-digit numbers comparison (AC) tasks showed a rather poor Cronbach’s alpha, probably because, in the former, participants not only processed the numerical dimension of the stimuli but also other physical dimensions (such as the convex hull, the area of the dots and so on) that are congruent or incongruent with the numerical dimension (see for instance [70]) and, in the latter, because of a ceiling effect, errors appeared mostly at random and probably due to inattention. Globally, it seems that core number subtests should be considered with caution: their correlation with STM, or their ability to differentiate between TA and MD students, are observed only in some grades, mostly in younger grades for single-digit comparison and subitizing, and in older grades for dots comparisons. The failure to find significant results for the single-digit comparison task with the RT measure is quite surprising given that previous research has found robust differences using this type of task [71]. One possible explanation is that, in this battery, RTs were not corrected for possible differences in global processing speed. Further research should add to the battery a measure for controlling this RT.

### 4.2. The Memory Domain Measures

The memory domain includes tasks that, to be solved successfully, require the student to recall number words (numbers dictation task), access memorized sequences, such as the counting sequence (enumeration task), or know which number comes before (previous number task) or after (next number task) another and retrieve arithmetical facts (addition and multiplication facts retrieval tasks).

Although previous research has shown that the ordinal dimension of numbers is very important and that judging whether a triplet of numbers is presented in the correct order (ascending or descending) or not is a very good predictor of math performance [21], we failed to find a strong correlation between the next and previous number tasks and students’ global math score, or much difference between MD and TA students on these two tasks. Further research should test whether those results are due to the specific task used here.

Regarding enumeration, some previous research has found that eight-year-old students with MLD are slow on object enumerations [20]. However, this was not found by Schleifer and Landerl (2011) in students of grades 2 to 4 [18]. The task used here showed that the AC measure is more sensitive for the young students (grades 1–2), whereas RT is better for the older students because these students showed a near ceiling performance.

The numbers dictation task proved to be a very interesting task as AC was significantly associated with the STM nearly in all grades. Moreover, significant differences were observed between the TA and the MD students in all grades, suggesting that it is a sensitive measure for all grades, while RT is not.

The addition and multiplication facts retrieval tasks have been used as diagnostic criteria for developmental dyscalculia in some studies [20,53]. In this study, we found significantly lower performance in students with MD, especially lower AC in students of grade 1 and longer RTs in students of grades 4–6 for additions and for multiplications and lower AC and longer RTs in students of grades 4–6. So, depending on the school grade, some dependent measures are more sensitive than others.

### 4.3. The Reasoning Domain Measures

The reasoning domain includes tasks testing the understanding of the place value system (multidigit number comparison), the performance on mental calculations (mental calculations task), word-problem solving (word-problem task), the grasping of basic calculation principles (calculation principles task) and the ability to infer numerical patterns (numerical patterns task).

Although, the AC of the multidigit numbers comparison task showed a rather poor Cronbach’s alpha, possibly be due to the fact that the last items with decimal numbers were much more difficult than the preceding ones, this task, as well as all the other ones from that domain, showed significant correlations with STM in all grades and allowed distinguishing between TA and MD students. AC measures in all these subtests are thus considered a sensitive measure for all grades tested. It is interesting to note that our word-problem task was very pure in that reading was not mandatory (i.e., the problems were spoken out loud) and calculation was not required. Still, under these conditions, we found clear weakness in MD students. We can conclude that these students’ difficulties with word-problem solving cannot be fully accounted for by possible associated reading problems [72] or calculation problems [73]. Indeed, we clearly also need to consider these students’ weaknesses in understanding, conceptualizing and finding the appropriate calculation to solve the problem [74,75].

### 4.4. The Visual-Spatial Domain Measures

The visual-spatial domain includes tasks that require analyzing 2D (squares task) and 3D (building blocks task) geometrical figures and also number line estimation tasks (number line 0–100 and 0–1000 tasks).

The building blocks task showed lower performance in students with MD in four grades (grades 1 and 4 to 6) and also significant correlations with the STM in four grades also (1, 3, 4 and 6). All the other subtests of this domain (the two number lines and the squares) showed, in all grades tested, significant correlations with STM and significant difference between TA and MD students. The poor internal consistency found in the squares task could be due to the variability of the processes required to solve the items (shapes discrimination, mental rotation and displacement). Therefore, further investigation of this task by analyzing students’ responses qualitatively should be conducted.

### 4.5. The Difficulties in Children with MD

Using this battery also brought new information on the difficulties encountered in MD. Indeed, one of the main characteristics of the children with MD is related to their arithmetical facts retrieval impairment [22,76] and this has even been used as a diagnostic criterion for developmental dyscalculia [20,53]. In this study, we also found arithmetical facts difficulties in students with MD, as in previous studies; however, we found even greater difficulties in solving multidigit calculations, understanding the properties and relations between the arithmetical operations themselves (calculation principles task) and matching a verbal situation (or a word problem) to an arithmetical operation. This suggests that diagnosing MD or dyscalculia on the simple basis of a test of arithmetic facts retrieval can be quite misleading.

Second, cognitive research has largely been considering the core number deficit present in MD children and has even hypothesized that such deficits would be the possible cause of dyscalculia. Here, although we observed some difficulties in these core number tasks in children with MD, these were rather small and did not concern all the grades. For instance, the weakness observed in subitizing did not remain beyond grade 2. The poor performances observed in the dots comparison task were observed in the older children (grades 4 to 6) but not in the younger ones [77] and, for the single-digit comparison, moderate differences were observed between MD and TA children in grades 1 and 2 only. Similarly, we failed to find a strong difficulty in the ordinal number processing (next and previous task) in MD children; we only found lower AC in the previous number task in grade 2. These findings suggest that the core number deficit might not be as strong as hypothesized and probably corresponds to a developmental delay.

By contrast, we found clear evidence for long-lasting difficulties in several other tasks. First, our data replicated the finding that children with MD are performing weakly on number line tasks [78] in all the grades considered. Furthermore, we also observed difficulties in two new tasks that required visual spatial processing, i.e., the squares and the building block tasks. The difficulties in visual analyses and visual imagery processes in MD should thus deserve further attention. Additionally, difficulties in number transcoding in children with MD were also observed. This is quite interesting as this process has mostly been studied in typically developing children and only a few case studies of children with MLD showing dysfunctions in this process were published [79,80].

In summary, we found that MD’s difficulties were mostly not affecting the core number tasks but, in fact, they concerned the arithmetical domains well beyond the arithmetical facts retrieval, number transcoding, number positioning on a line and other visual-spatial tasks, such as the squares and the blocks. This may open new avenues of research in the field of mathematical learning difficulties.

### 4.6. Limitations and Future Perspectives

This work still has some limitations. In particular, we assessed a large sample of students, but we did not have personal information regarding each participant (such as the precise age, the level of reading proficiency, the IQ, the socio-economic level of the parents, etc.). Further research should take such variables into account.

The validation of an assessment tool is work that is always in progress. Future research should examine the internal structure of the battery by running factorial analyses for each grade and measure the test-retest fidelity of the individual measures. This will allow further examination of the extent to which each subtest really elicits skills from the indented domain. We could also measure the sensitivity and specificity of the battery by comparing typically developing students with MLD students or students who have received a diagnosis of dyscalculia. Moreover, error analysis on the squares task and the administration of that task to grade 1 and grade 2 students could help to better understand the processes involved in solving it. Finally, controlling the reaction time for the single-digit numbers comparison task should be implemented to increase the sensitivity of this measure.

## 5. Conclusions

As a whole, the first results of the MathPro Test administration support the reliability and the validity of the tool. Children’s performance was sensitive to the level of difficulty of the items and to the school grade. Among the different subtests, those from the reasoning domain (the multidigit numbers comparison, the mental calculations, the word problems, the calculation principles and the numerical patterns), the visuospatial domain (the two number lines, the squares and, more moderately, the blocks) and some of the memory domains (numbers dictation) seem to be the most sensitive for the age range tested. Other subtests seem to be interesting for younger students only (subitizing, single-digit comparisons (RT), enumeration (AC) and addition fact retrieval (AC)), while others are interesting specifically for older students (dots comparison, addition facts retrieval (RT) and multiplication facts retrieval (AC and RT)). These results also suggest that using core number skills or simply arithmetic facts retrieval to identify children at risk of difficulties in mathematics is not enough.

To conclude, the MathPro Test seems to be a very promising tool that allows having an exceptionally broad assessment of the numerical processes for large scale research and for clinicians to sketch out the mathematical profile of MD children or those with dyscalculia.

## Figures and Tables

**Figure 1 behavsci-10-00126-f001:**
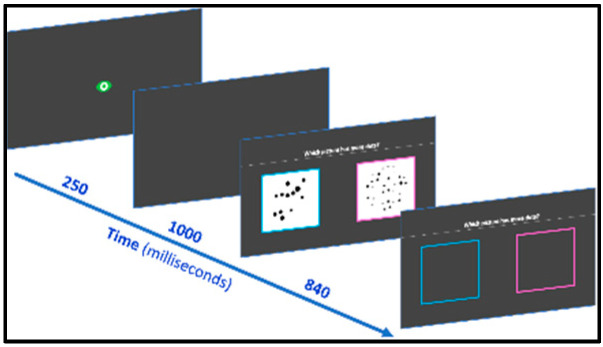
Dots magnitude comparison subtest screenshots.

**Figure 2 behavsci-10-00126-f002:**
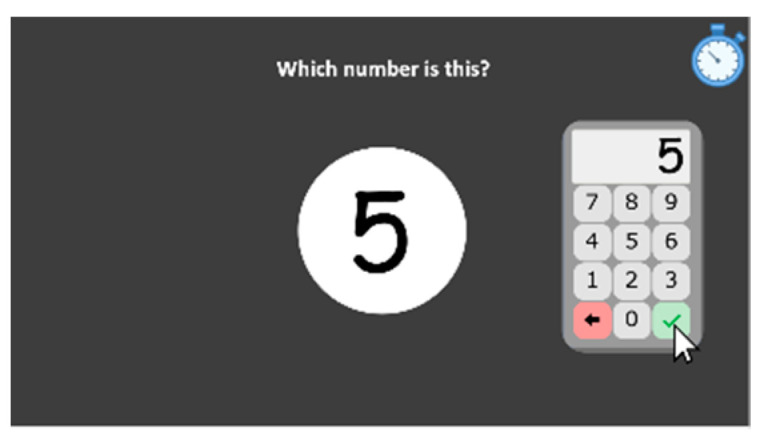
Screen keyboard use subtest screenshot.

**Figure 3 behavsci-10-00126-f003:**
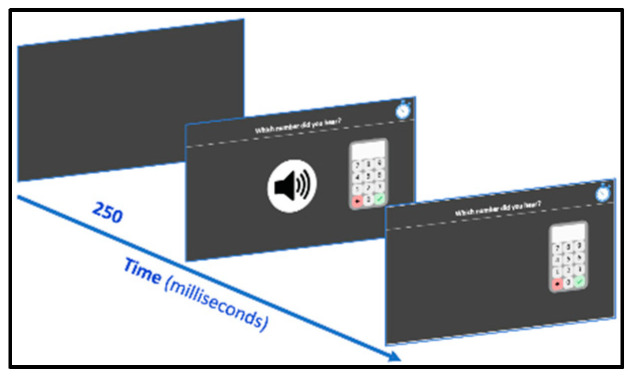
Numbers dictation subtest screenshots.

**Figure 4 behavsci-10-00126-f004:**
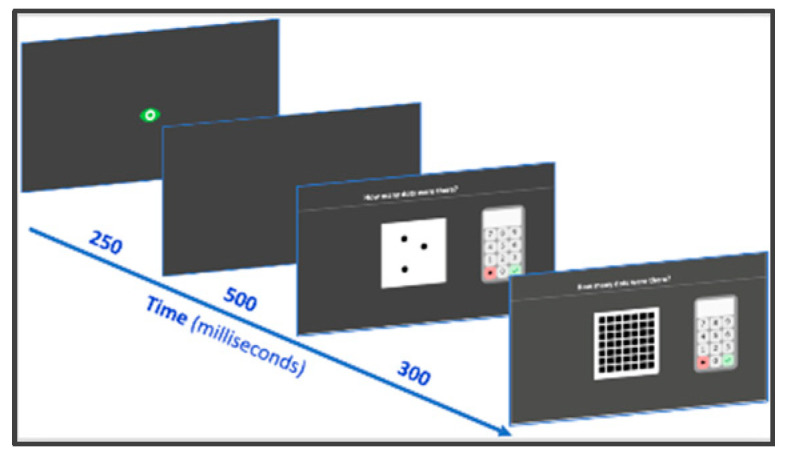
Subitizing subtest screenshots.

**Figure 5 behavsci-10-00126-f005:**
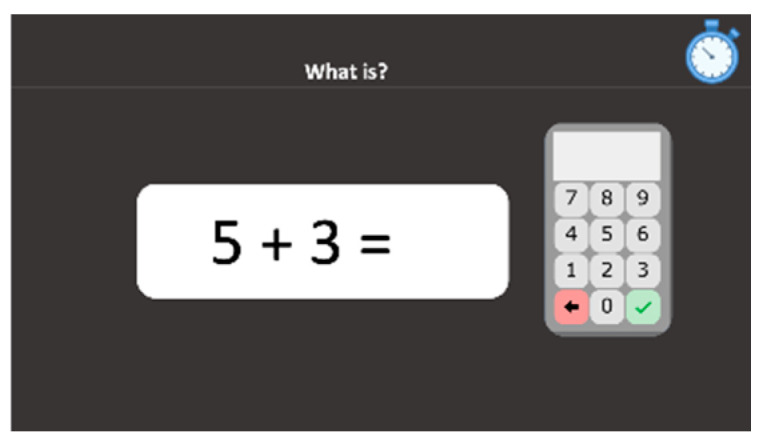
Addition facts retrieval subtest screenshot.

**Figure 6 behavsci-10-00126-f006:**
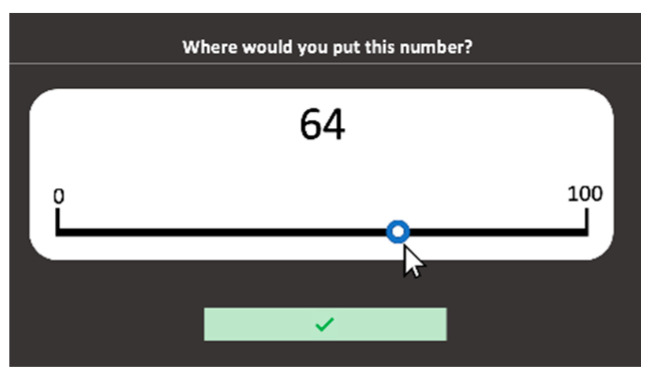
Number lines 0–100 subtest screenshot.

**Figure 7 behavsci-10-00126-f007:**
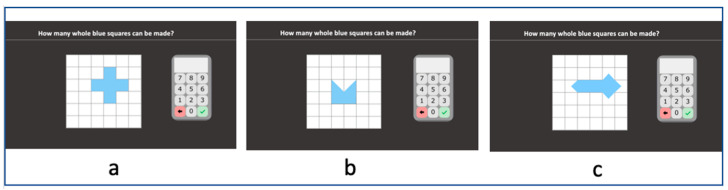
Squares subtest screenshots. (**a**) contain only whole squares; (**b**) contain a combination of whole squares and half-square triangles; (**c**) quarter-square triangles are also included.

**Figure 8 behavsci-10-00126-f008:**
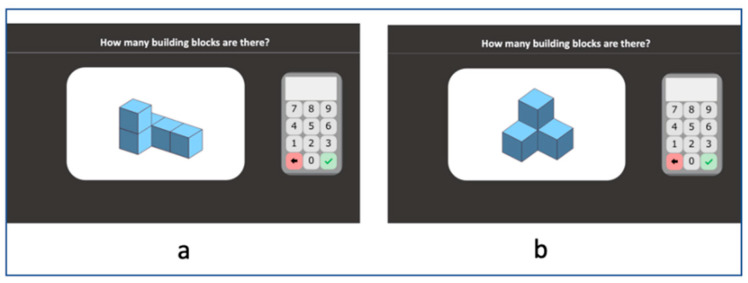
Building blocks subtest screenshots. (**a**) stimulus example a; (**a**) stimulus example b.

**Figure 9 behavsci-10-00126-f009:**
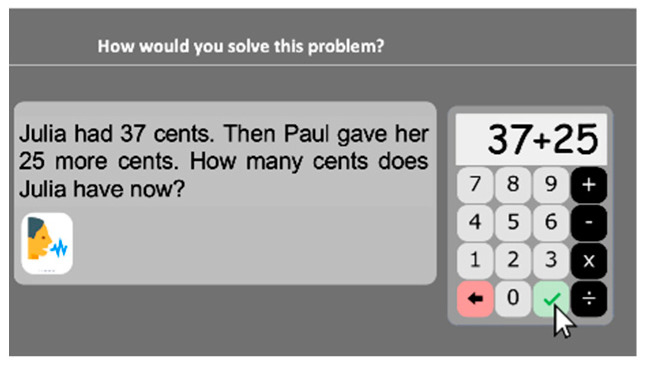
Word problems subtest screenshot.

**Figure 10 behavsci-10-00126-f010:**
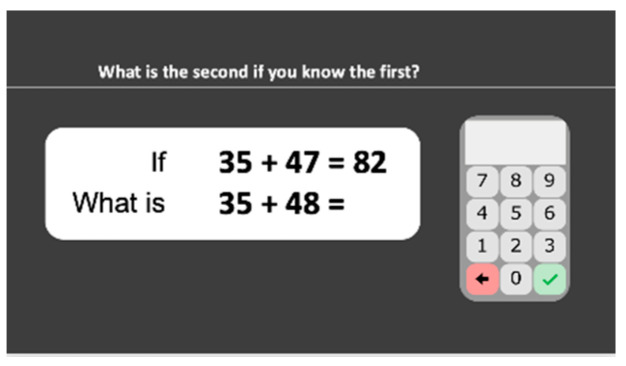
Calculation principles subtest screenshot.

**Figure 11 behavsci-10-00126-f011:**
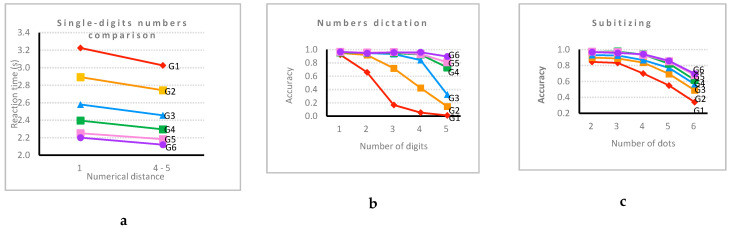
Difficulty level × grade interaction on (**a**) single-digits number comparison, (**b**) number dictation, (**c**) subitizing, (**d**) building blocks, (**e**) numerical patterns and (**f**) number lines.

**Table 1 behavsci-10-00126-t001:** Descriptive statistics of the sample.

Grade	Sex		Sum
	Boys	Girls	
1	50	56	106
2	55	43	98
3	52	50	102
4	55	56	111
5	52	40	92
6	56	57	113
**Sum**	320	302	622

**Table 2 behavsci-10-00126-t002:** Mathematical Profile Test (MathPro Test) subtests for each domain.

Domain			
Core Number	Memory	Reasoning	Visual-Spatial
1. Dots comparison 2. Single-digit numbers comparison 8. Subitizing	5. Numbers dictation 6. Νext number 7. Previous number 9. Enumeration 10. Addition facts retrieval 11. Multiplication facts retrieval	3. Multidigit numbers comparison 12. Mental calculations 17. Word problems 18. Calculation principles 19. Numerical patterns	13. Number lines 0–100 14. Number lines 0–1000 15. Squares 16. Building blocks

Note: subtest “4. Screen keyboard use” was not included in the table because it was introduced only for controlling individuals’ processing speed.

**Table 3 behavsci-10-00126-t003:** Number of items of MathPro subtests administrated per grade.

Subtests	Grade 1	Grade 2	Grade 3	Grades 4–6
1. Dots comparison	30	30	30	30
2. Single-digit numbers comparison	24	24	24	24
3. Multidigit numbers comparison	0	6	9	12
4. Screen keyboard use	10	10	10	10
5. Numbers dictation *	30	30	30	30
6. Next number *	12	18	18	18
7. Previous number *	12	18	18	18
8. Subitizing	20	20	20	20
9. Enumeration	14	14	14	14
10. Addition facts retrieval	12	12	12	12
11. Multiplication facts retrieval	0	0	14	14
12. Mental calculations	0	12	24	24
13. Number lines 0–100	22	22	22	22
14. Number lines 0–1000	0	0	22	22
15. Squares	0	0	10	10
16. Building blocks	8	8	8	8
17. Word problems *	0	10	18	18
18. Calculation principles *	0	0	15	15
19. Numerical patterns *	18	18	18	18

* Subtest terminates after 3 consecutive errors.

**Table 4 behavsci-10-00126-t004:** Descriptive statistics of MathPro subtests per grade.

Individual Measures	Grade 1		Grade 2		Grade 3		Grade 4		Grade 5		Grade 6	
	*M*	*SD*	*M*	*SD*	*M*	*SD*	*M*	*SD*	*M*	*SD*	*M*	*SD*
Dots comparison (AC)	0.55	0.13	0.60	0.10	0.62	0.10	0.63	0.09	0.69	0.08	0.67	0.09
Single-digit numbers comparison (AC)	0.95	0.06	0.97	0.39	0.98	0.05	0.99	0.03	0.98	0.05	0.98	0.04
Single-digit numbers comparison (RT)	3113	437	2799	385	2515	354	2337	205	2211	314	2153	187
Multidigit numbers comparison (AC)	-	-	0.85	0.18	0.88	0.14	0.80	0.13	0.87	0.12	0.90	0.12
Multidigit numbers comparison (RT)	-	-	4027	944	3512	503	3075	360	2903	444	2749	285
Screen keyboard use (AC)	0.96	0.01	0.98	0.01	0.98	0.01	0.99	0.01	0.99	0.01	0.99	0.01
Screen keyboard use (RT)	4234	792	3621	523	3204	399	2850	275	2682	350	2484	259
Numbers dictation (AC)	0.36	0.14	0.63	0.20	0.79	0.17	0.90	0.16	0.93	0.12	0.95	0.12
Numbers dictation (RT)	3670	1339	3890	862	2487	491	2679	600	2539	438	2243	404
Next number (AC)	0.50	0.35	0.64	0.28	0.83	0.25	0.91	0.09	0.92	0.02	0.95	0.12
Next number (RT)	5682	1779	5513	1384	3605	889	3569	596	3169	519	2772	370
Previous number (AC)	0.53	0.37	0.73	0.24	0.87	0.21	0.94	0.12	0.95	0.14	0.97	0.07
Previous number (RT)	5942	2079	5189	1115	3259	610	3244	510	2957	439	2618	347
Subitizing (AC)	0.64	0.20	0.74	0.16	0.79	0.14	0.85	0.11	0.87	0.11	0.87	0.12
Enumeration (AC)	0.71	0.29	0.81	0.23	0.85	0.18	0.92	0.13	0.92	0.13	0.89	0.15
Enumeration (RT)	11830	6553	8387	1850	7529	1550	6623	1263	6115	1250	5431	1024
Addition facts retrieval (AC)	0.82	0.25	0.91	0.19	0.94	0.14	0.98	0.04	0.98	0.04	0.96	0.13
Addition facts retrieval (RT)	8178	4130	5402	1847	4343	2617	3343	778	3030	606	2568	428
Multiplication facts retrieval (AC)	-	-	-	-	0.80	0.24	0.92	0.14	0.95	0.058	0.92	0.16
Multiplication facts retrieval (RT)	-	-	-	-	8550	3598	6186	2214	4992	1539	4333	2190
Mental calculations (AC)	-	-	0.53	0.27	0.63	0.23	0.80	0.19	0.85	0.14	0.84	0.15
Number lines 0–100 (PAE)	17.24	8.12	8.42	4.29	6.48	2.81	5.14	1.62	4.73	1.49	4.59	1.68
Number lines 0–1000 (PAE)	-	-	-	-	11.02	6.49	7.75	4.42	5.97	3.08	5.37	2.99
Squares (AC)	-	-	-	-	0.59	0.23	0.68	0.17	0.73	0.16	0.77	0.16
Building blocks (AC)	0.58	0.27	0.78	0.26	0.81	0.23	0.86	0.19	0.91	0.14	0.93	0.12
Word problems (AC)	-	-	0.44	0.30	0.36	0.24	0.54	0.29	0.68	0.31	0.81	0.18
Calculation principles (AC)	-	-	-	-	0.22	0.22	0.33	0.23	0.36	0.18	0.51	0.28
Numerical patterns (AC)	0.21	0.16	0.38	0.18	0.45	0.17	0.52	0.20	0.58	0.21	0.60	0.20

Note: reaction times (RTs) calculated for the items with the accuracy leading to a ceiling (>0.85).

**Table 5 behavsci-10-00126-t005:** Cronbach’s a coefficient for all the measures.

Measure	Cronbach’s α	Internal Consistency Level
Numbers dictation (AC)	0.95	Good or excellent
Screen keyboard use (RT)	0.94	
Calculation principles (AC)	0.92	
Number lines 0–100 (PAE)	0.92	
Next number (AC)	0.87–0.93	
Word problems (AC)	0.86–0.93	
Previous number (AC)	0.84–0.93	
Single-digit numbers comparison (RT)	0.82–0.93	
Multidigit numbers comparison (RT)	0.81–0.90	
Numerical patterns (AC)	0.89	
Numbers dictation (RT)	0.56–0.89	
Enumeration (RT)	0.88	
Addition facts retrieval (AC)	0.88	
Number lines 0–1000 (PAE)	0.88	
Multiplication facts retrieval (AC)	0.86	
Mental calculations (AC)	0.85–0.87	
Multiplication facts retrieval (RT)	0.85	
Enumeration (AC)	0.85	
Addition facts retrieval (RT)	0.84	
Building blocks (AC)	0.79	Acceptable
Subitizing (AC)	0.78	
Next number (RT)	0.66–0.78	
Previous number (RT)	0.57–0.74	
Squares (AC)	0.66	Poor
Screen calculator use (AC)	0.62	
Single-digit numbers comparison (AC)	0.57	
Multidigit numbers comparison (AC)	0.44–0.55	
Dots comparison (AC)	0.42	Unacceptable

**Table 6 behavsci-10-00126-t006:** Results of the ANOVAs for grade and difficulty level of MathPro subtests.

	Grade	η^2^	Difficulty	η^2^	Grade × Difficulty	η^2^
Dots comparison (AC)	F_5,541_ = 15.93 ***	0.13	^1^ F_3.678,1989_ = 102.10 ***	0.09	*p* = 0.062	
Single-digit numbers comparison (AC)	F_5,615_ = 10.72 ***	0.08	F_1,615_ = 67.37 ***	0.18	*p* = 0.298	
Single-digit numbers comparison (RT)	F_5,615_ = 136.60 ***	0.53	F_1,615_ = 252.30 ***	0.29	F_5,615_ = 7.11 ***	0.06
Multidigit numbers comparison ^4^ (AC)	F_2,312_ = 17.30 ***	0.10	^2^ F_31.664,519_ = 206.34 ***	0.40	^2^ F_3.328,519_ = 23.45 ***	0.13
Multidigit numbers comparison ^4^ (RT)	F_2,311_ = 21.72 ***	0.12	^1^ F_1.775,552_ = 63.06 ***	0.17	*p* = 0.407	
Numbers dictation (AC)	F_5,614_ = 242.84 ***	0.66	^1^ F_3.234,1986_ = 654.23 ***	0.52	^1^ F_16.17,1986_ = 107.85 ***	0.47
Numbers dictation ^4^ (RT)	F_2,300_ = 27.54 ***	0.16	^1^ F_2.652,796_ = 4.74 **	0.02	^1^ F_5.304,796_ = 6.03 ***	0.04
Next number ^3^ (AC)	F_3,413_ = 8.72 ***	0.06	^1^ F_1.781,734_ = 37.84 **	0.08	^1^ F_5.343,734_ = 5.08 **	0.04
Next number ^4^ (RT)	F_2,304_ = 20.01 ***	0.17	*p* = 0.972		F_2,304_ = 5.49 **	0.04
Previous number ^3^ (AC)	F_3,413_ = 9.72 ***	0.07	^1^ F_1.700,702_ = 34.55 **	0.08	^1^ F_5.101,702_ = 7.06 *	0.05
Previous number ^4^ (RT)	F_2,306_ = 18.64 ***	0.11	*p* = 0.516		F_2,306_ = 9.93 **	0.06
Subitizing (AC)	F_5,614_ = 41.622 ***	0.25	^2^ F_2.802,1721_ = 458.88 ***	0.43	^2^ F_14.012,1721_ = 4.984 ***	0.04
Enumeration (AC)	F_5,616_ = 17.67 ***	0.13	F_1,616_ = 74.64 ***	0.11	F_5,616_ = 2.60 *	0.02
Enumeration ^5^ (RT)	F_4,500_ = 8.74 ***	0.07	*p* = 0.216		*p* = 0.584	
Addition facts retrieval (AC)	F_5,615_ = 15.36 ***	0.11	*p* = 0.219		*p* = 0.099	
Addition facts retrieval ^5^ (RT)	F_4,510_ = 5.32 ***	0.04	*p* = 0.991		*p* = 0.773	
Multiplication facts retrieval ^3^ (AC)	F_3,412_ = 15.95 ***	0.10	F_1,412_ = 13.02 ***	0.03	*p* = 0.783	
Multiplication facts retrieval ^4^ (RT)	F_2,412_ = 19.49 ***	0.13	*p* = 0.799		*p* = 0.087	
Mental calculations (AC)	F_3,413_ = 31.64 ***	0.19	^1^ F_1.745,721_ = 219.09 ***	0.35	^1^ F_5.236,721_ = 8.34 ***	0.06
Number line tasks ^6^ (PAE)	F_5,611_ = 147.55 ***	0.55				
Squares (AC)	F_3,413_ = 33.04 ***	0.20				
Building blocks (AC)	F_3,412_ = 21.75 ***	0.14	F_1,412_ = 1216.52 ***	0.75	F_3,412_ = 9.84 ***	0.07
Word problems ^3^ (AC)	F_5,610_ = 37.71 ***	0.24	F_1,610_ = 230.76 ***	0.27	F_5,610_ = 6.86 ***	0.05
Calculation principles (AC)	F_3,403_ = 61.69 ***	0.32	F_2,806_ = 335.39 ***	0.45	F_6,806_ = 9.07 ***	0.06
Numerical patterns (AC)	F_3,403_ = 25.23 ***	0.16	^1^ F_1.791,722_ = 511.78 ***	0.60	^1^ F_5.374,722_ = 15.36 ***	0.10

*** *p* < 0.001; ** *p* < 0.01; * *p* < 0.05. AC = accuracy; RT = reaction time (in milliseconds); PAE = percentage of absolute errors; ns = nonsignificant. ^1^ Huynh–Feldt correction; ^2^ Greenhouse–Geisser correction; ^3^ grades 3–6; ^4^ grades 4–6; ^5^ grades 2–6; ^6^ includes both number line 0–100 and 0–1000 tasks.

**Table 7 behavsci-10-00126-t007:** Person correlation coefficients between the MathPro index score and the standardized math test.

Grade	1	2	3	4	5	6
Dots comparison (AC)	0.124	0.278 **	0.299 **	0.201 *	0.009	0.052
Single-digit numbers comparison (AC)	0.087	0.097	0.100	0.010	0.107	0.187
Single-digit numbers comparison (RT)	0.257 **	0.022	0.020	0.051	0.009	0.093
Multidigit numbers comparison (AC)	-	0.385 **	0.368 **	0.538 **	0.289 **	0.192 *
Multidigit numbers comparison (RT)	-	0.181	0.252 *	0.158	0.166	0.075
Numbers dictation (AC)	0.283 **	0.439 **	0.403 **	0.206*	0.307 **	0.148
Numbers dictation (RT)	-	-	0.035	0.169	0.023	0.036
Next number (AC)	0.256 **	0.305 **	0.094	0.106	0.187	0.042
Next number (RT)	-	-	0.057	0.137	0.182	0.024
Previous number (AC)	0.303 **	0.478 **	0.109	0.167	0.141	0.086
Previous number (RT)			0.100	0.028	0.072	0.042
Subitizing (AC)	0.288 **	0.263 **	0.152	0.013	0.184	0.095
Enumeration (AC)	0.382 **	0.249 *	0.014	0.152	0.126	0.154
Enumeration (RT)	-	-	0.235 *	0.017	0.179	0.184
Addition facts retrieval (AC)	0.434 **	0.260 **	0.162	0.313 **	0.006	0.034
Addition facts retrieval (RT)	-	0.315 **	0.180	0.287 **	0.373 **	0.288 **
Multiplication facts retrieval (AC)	-	-	0.278 **	0.336 **	0.008	0.062
Multiplication facts retrieval (RT)	-	-	-	0.161	0.433 **	0.192 *
Mental calculations (AC)		0.573 **	0.439 **	0.495 **	0.460 **	0.325 **
Number lines 0–100 (PAE)	0.374 **	0.476 **	0.430 **	0.292 **	0.434 **	0.274 **
Number lines 0–1000 (PAE)	-	-	0.569 **	0.488 **	0.414 **	0.394 **
Squares (AC)	-	-	0.365 **	0.433 **	0.277 **	0.319 **
Building blocks (AC)	0.446 **	0.185	0.219*	0.327 **	0.004	0.196 **
Word problems (AC)	-	0.518 **	0.391 **	0.613 **	0.335 **	0.495 **
Calculation principles (AC)	-	-	0.364 **	0.425 **	0.411 **	0.189
Numerical patterns (AC)	0.521 **	0.477 **	0.298 **	0.509 **	0.462 **	0.475 **
MathPro index score	0.595 **	0.604 **	0.568 **	0.627 **	0.637 **	0.472 **

* *p* < 0.05; ** *p* < 0.01.

**Table 8 behavsci-10-00126-t008:** MANOVA and MANCOVA for differences in means between TA and MD children for grade 1, 2, 3.

Individual Measures	Grade 1				Grade 2				Grade 3			
	TA	MD	F_1,103_	Sig	TA	MD	F_1,91_	Sig	TA	MD	F_1,93_	Sig
	*n* = 20	*n* = 83			*n* = 75	*n* = 16			*n* = 73	*n* = 20		
	*M (SD)*	*M (SD)*			*M (SD)*	*M (SD)*			*M (SD)*	*M (SD)*		
Dots comparison (AC)	0.55 (0.13)	0.54 (0.11)	0.047	0.829	0.60 (0.01)	0.57 (0.01)	1.30	0.256	0.63 (0.01)	0.60 (0.01)	1.152	0.286
Single-digit numbers comparison (AC)	0.95 (0.07)	0.94 (0.05)	0.077	0.782	0.98 (0.03)	0.95 (0.06)	9.19	0.003	0.98 (0.04)	0.98 (0.03)	0.305	0.582
Single-digit numbers comparison (RT)	3324 (390)	3050 (421)	7.286	0.008	2788 (493)	2807 (376)	0.385	0.537	2490 (367)	2583 (380)	0.132	0.717
Multidigit numbers comparison (AC)	-	-	-	-	0.88 (0.17)	0.71 (0.21)	11.43	0.001	0.92 (0.12)	0.78 (0.15)	17.318	<0.001
Multidigit numbers comparison (RT)	-	-	-	-	4229 (1052)	3991 (920)	1.044	0.310	3421 (464)	3832 (596)	10.036	0.002
Numbers dictation (AC)	0.38 (0.13)	0.31 (0.15)	4.34	0.040	0.66 (0.21)	0.53 (0.18)	5.47	0.022	0.83 (0.14)	0.69 (0.19)	12.198	0.001
Numbers dictation (RT)	-	-	-	-	-	-	-	-	3242 (610)	3407 (746)	0.175	0.676
Next number (AC)	0.53 (0.34)	0.39 (0.34)	2.76	0.100	0.65 (0.29)	0.63 (0.22)	0.058	0.810	0.83 (0.03)	0.82 (0.03)	0.029	0.866
Next number (RT)	-	-	-	-	-	-	-	-	3572 (800)	3838 (1268)	0.526	0.470
Previous number (AC)	0.56 (0.35)	0.44 (0.40)	1.84	0.179	0.76 (0.23)	0.60 (0.27)	6.13	0.015	0.87 (0.20)	0.82 (0.28)	0.775	0.381
Previous number (RT)	-	-	-	-	-	-	-	-	3239 (644)	3361 (581)	0.143	0.706
Subitizing	0.66 (0.20)	0.54 (0.18)	6.32	0.014	0.75 (0.13)	0.66 (0.24)	5.02	0.028	0.79 (0.12)	0.78 (0.12)	0.236	0.628
Enumeration (AC)	0.76 (0.24)	0.52 (0.32)	14.34	<0.001	0.82 (0.22)	0.75 (0.29)	1.11	0.295	0.84 (0.18)	0.85 (0.21)	0.016	0.901
Enumeration (RT)	-	-	-	-	-	-	-	-	7431 (1547)	7874 (1847)	0.607	0.438
Addition facts retrieval (AC)	0.88 (0.18)	0.66 (0.34)	15.31	<0.001	0.92 (0.19)	0.85 (0.25)	1.55	0.216	0.94 (0.13)	0.95 (0.08)	0.075	0.784
Addition facts retrieval (RT)	-	-	-	-	5177 (1667)	6816 (2390)	12.113	0.001	4096 (1428)	5444 (5187)	3.347	0.071
Multiplication facts retrieval (AC)	-	-	-	-	-	-	-	-	0.82 (0.22)	0.75 (0.25)	1.320	0.254
Mental calculations (AC)	-	-	-	-	0.59 (0.25)	0.30 (0.25)	17.52	<0.001	0.66 (0.23)	0.52 (0.18)	5.408	0.022
Number lines 0–100 (PAE)	15.67 (7.46)	23.21 (8.37)	15.51	<0.001	7.7 (3.88)	11.24 (4.65)	10.85	0.002	5.77 (1.79)	8.92 (4.40)	22.374	<0.001
Number lines 0–1000 (PAE)	-	-	-	-	-	-	-	-	8.82 (4.81)	16.13 (6.69)	27.984	<0.001
Squares (AC)	-	-	-	-	-	-	-	-	0.62 (0.21)	0.45 (0.23)	9.514	0.003
Building blocks (AC)	0.63 (0.25)	0.46 (0.29)	7.39	0.008	0.79 (0.27)	0.72 (0.26)	0.82	0.368	0.83 (0.23)	0.77 (0.18)	1.029	0.313
Word problems (AC)	-	-	-	-	0.49 (0.30)	0.23 (0.26)	10.85	0.001	0.40 (0.25)	0.28 (0.16)	3.743	0.041
Calculation principles (AC)	-	-	-	-	-	-	-	-	0.26 (0.24)	0.09 (0.09)	7.801	0.006
Numerical patterns (AC)	0.25 (0.15)	0.08 (0.09)	21.95	<0.001	0.42 (0.18)	0.28 (0.16)	6.78	0.011	0.46 (0.17)	0.41 (0.15)	1.240	0.268

Note: TA = typically achieving; MD =difficulties in mathematics.

**Table 9 behavsci-10-00126-t009:** MANOVA and MANCOVA for differences in means between TA and MD children for grades 4–6.

Individual Measures	TA	MD	F_1,278_	Sig
	*n* = 212	*n* = 76		
	*M (SD)*	*M (SD)*		
Dots comparison (AC)	0.67 (0.09)	0.64 (0.09)	5.287	0.022
Single-digit numbers comparison (AC)	0.99 (0.04)	0.99 (0.03)	0.103	0.749
Single-digit numbers comparison (RT)	2239 (244)	2226 (290)	3.466	0.064
Multidigit numbers comparison (AC)	0.88 (0.11)	0.79 (0.14)	28.790	<0.001
Multidigit numbers comparison (RT)	2885 (352)	3009 (478)	4.31	0.039
Numbers dictation (AC)	0.94 (0.12)	0.87 (0.17)	14.752	<0.001
Numbers dictation (RT)	4079 (593)	4211 (811)	0.694	0.405
Next number (AC)	0.93 (0.11)	0.91 (0.17)	2.395	0.123
Next number (RT)	3125 (543)	3310 (724)	3.653	0.057
Previous number (AC)	0.96 (0.11)	0.95 (0.07)	0.591	0.443
Previous number (RT)	2936 (488)	2961 (578)	0.333	0.564
Subitizing (AC)	0.87 (0.11)	0.85 (0.12)	3.725	0.055
Enumeration (AC)	0.92 (0.11)	0.90 (0.16)	1.949	0.164
Enumeration (RT)	5935 (1209)	6392 (1480)	5.238	0.023
Addition facts retrieval (AC)	0.98 (0.05)	0.96 (0.06)	1.073	0.301
Addition facts retrieval (RT)	2878 (598)	3237 (902)	14.173	<0.001
Multiplication facts retrieval (AC)	0.96 (0.08)	0.90 (0.17)	7.244	0.008
Multiplication facts retrieval (RT)	4990 (2215)	5827 (2165)	6.403	0.012
Mental calculations (AC)	0.87 (0.12)	0.72 (0.21)	54.627	<0.001
Number lines 0–100 (PAE)	4.50 (1.33)	5.68 (2.06)	30.571	<0.001
Number lines 0–1000 (PAE)	5.42 (2.93)	9.43 (4.58)	70.988	<0.001
Squares (AC)	0.77 (0.16)	0.66 (0.18)	24.387	<0.001
Building blocks (AC)	0.92 (0.14)	0.86 (0.19)	8.479	0.004
Word problems (AC)	0.75 (0.24)	0.48 (0.28)	60.986	<0.001
Calculation principles (AC)	0.44 (0.24)	0.31 (0.24)	13.964	<0.001
Numerical patterns (AC)	0.62 (0.20)	0.47 (0.16)	30.403	<0.001

**Table 10 behavsci-10-00126-t010:** Sensitivity level of individual measures on each task.

Domain	Subtests	Criterion	Grade 1	Grade 2	Grade 3	Grade 4	Grade 5	Grade 6
**Core**	Dots comparison (AC) ^α^	STMC		✓	✓	✓		
		MD-TA				✓	✓	✓
	Single-digit numbers comparison (AC) ^α^	STMC						
		MDC		✓				
	Single-digit numbers comparison (RT)	STMC	✓					
		MD-TA	✓					
	Subitizing	STMC	✓	✓				
		MD-TA	✓	✓				
**Memory**	Numbers dictation (AC)	STMC	✓	✓	✓	✓	✓	
		MD-TA	✓	✓	✓	✓	✓	✓
	Numbers dictation (RT) ^α^	STMC						
		MD-TA						
	Next number (AC)	STMC	✓	✓				
		MD-TA						
	Next number (RT) ^α^	STMC						
		MD-TA						
	Previous number (AC)	STMC	✓	✓				
		MD-TA						
	Previous number (RT) ^α^	STMC						
		MD-TA						
	Enumeration (AC)	STMC	✓	✓				
		MD-TA	✓					
	Enumeration (RT)	STMC			✓			
		MD-TA				✓	✓	✓
	Addition facts retrieval (AC)	STMC	✓	✓		✓		
		MD-TA	✓					
	Addition facts retrieval (RT)	STMC		✓		✓	✓	✓
		MD-TA		✓		✓	✓	✓
	Multiplication facts retrieval (AC)	STMC			✓	✓		
		MD-TA				✓	✓	✓
	Multiplication facts retrieval (RT)	STMC					✓	✓
		MD-TA				✓	✓	✓
**Reasoning**	Multidigit numbers comparison (AC) ^α^	STMC		✓	✓	✓	✓	✓
		MD-TA		✓	✓	✓	✓	✓
	Multidigit numbers comparison (RT)	STMC			✓			
		MD-TA			✓	✓	✓	✓
	Mental calculations	STMC		✓	✓	✓	✓	✓
		MD-TA		✓	✓	✓	✓	✓
	Word problems	STMC		✓	✓	✓	✓	✓
		MD-TA		✓	✓	✓	✓	✓
	Calculation principles	STMC			✓	✓	✓	
		MD-TA			✓	✓	✓	✓
	Numerical patterns	STMC	✓	✓	✓	✓	✓	✓
		MD-TA	✓	✓		✓	✓	✓
**Visual-spatial**	Number lines 0–100	STMC	✓	✓	✓	✓	✓	✓
		MD-TA	✓	✓	✓	✓	✓	✓
	Number lines 0–1000	STMC			✓	✓	✓	✓
		MD-TA			✓	✓	✓	✓
	Squares ^α^	STMC			✓	✓	✓	✓
		MD-TA			✓	✓	✓	✓
	Building blocks	STMC	✓		✓	✓		✓
		MD-TA	✓			✓	✓	✓

Note: α = indicates poor Cronbach’s a, STMC = significant correlation with STM and MD-TA = significant difference between MD and TA students. The grid format indicates that the task is not given or the measure was not included at the analysis.

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
