# Peer review of "Mathematical Profile Test: A Preliminary Evaluation of an Online Assessment for Mathematics Skills of Children in Grades 1–6"

_behavsci, 2020, doi:10.3390/bs10080126_

Round 1
Reviewer 1 Report
This paper addresses a good quality piece of research. Basically, it presents the raising up of a tool for measuring mathematical competence within children from 1-6 Grades. It is quite interesting because this novel tool is compared with already-known instruments (such as STM) and it seems to be consistent and strong enough. Several advantanges are equally showed, so the development of this MathPro test is, as authors stated out in the abstract, promising.
I recommend the publication of this article with minor revisions, such as the ones I suggest in the following lines:
Line 1, page 6. Helsinki ethics declaration, please referece it.
Line 24, page 6. Please revise, it seems to be an altenate table title.
Describing sub-test: perhaps it should be clarifying including some screenshots. I see they are in the Appendix B, but they could be also included (some of them) in the main text. Additionally, I would reduce the lenght of this part and a small discussion could be done by analyzing the objectives and focus these sub-test have got in Discussion Section.
Line 31, page 11: SPSS 25 has a User's manual. Please reference it as SPSS bibliographical cite.
Results section: An overwhelming data set is presented in this section in order to prove the internal consistency of the statistical process. I would try to resume them into one easy-to-read table, with qualitative descriptions of such consistency.
Author Response
Dear reviewer,
thank you for reviewing our paper.
Please find attached the response to your comments.
Sincerely,
The authors

Reviewer 2 Report
Please see attached file.

Author Response

(The authors gave the same response as above.)

Reviewer 3 Report
The paper presents developed tool for measuring math skills for primary children. Paper refers to all significant sources in the field and builds on the previous expertise of authors. The methodology was used correctly and the results confirm that the developed tool provides usable outcome.
The paper is well written and presents interesting results, I have only minor comments.
The statistical analysis is not described in sufficient deep in the Methodology.
The is a lot of formatting issues in the paper. The tables are difficult to read.
The reference 48 is divided into the two lines.
Author Response

(The authors gave the same response as above.)
